# Region-specific Diffeomorphic Metric Mapping

**Zhengyang Shen**
UNC Chapel Hill
zyshen@cs.unc.edu

**François-Xavier Vialard**
LIGM, UPEM
francois-xavier.vialard@u-pem.fr

**Marc Niethammer**
UNC Chapel Hill
mn@cs.unc.edu

## Abstract

We introduce a region-specific diffeomorphic metric mapping (RDMM) registration approach. RDMM is non-parametric, estimating spatio-temporal velocity fields which parameterize the sought-for spatial transformation. Regularization of these velocity fields is necessary. In contrast to existing non-parametric registration approaches using a fixed spatially-invariant regularization, for example, the large displacement diffeomorphic metric mapping (LDDMM) model, our approach allows for spatially-varying regularization which is advected via the estimated spatio-temporal velocity field. Hence, not only can our model capture large displacements, it does so with a spatio-temporal regularizer that keeps track of how regions deform, which is a more natural mathematical formulation. We explore a family of RDMM registration approaches: 1) a registration model where regions with separate regularizations are pre-defined (e.g., in an atlas space or for distinct foreground and background regions), 2) a registration model where a general spatially-varying regularizer is estimated, and 3) a registration model where the spatially-varying regularizer is obtained via an end-to-end trained deep learning (DL) model. We provide a variational derivation of RDMM, showing that the model can assure diffeomorphic transformations in the continuum, and that LDDMM is a particular instance of RDMM. To evaluate RDMM performance we experiment 1) on synthetic 2D data and 2) on two 3D datasets: knee magnetic resonance images (MRIs) of the Osteoarthritis Initiative (OAI) and computed tomography images (CT) of the lung. Results show that our framework achieves comparable performance to state-of-the-art image registration approaches, while providing additional information via a learned spatio-temporal regularizer. Further, our deep learning approach allows for very fast RDMM and LDDMM estimations. Code is available at https://github.com/uncbiag/registration.

## 1 Introduction

Quantitative analysis of medical images frequently requires the estimation of spatial correspondences, i.e.image registration. For example, one may be interested in capturing knee cartilage changes over time, localized changes of brain structures, or how organs at risk move between planning and treatment for radiation treatment. Specifically, image registration seeks to estimate the spatial transformation between a source image and a target image, subject to a chosen transformation model.

Transformations can be parameterized via low-dimensional parametric models (e.g., an affine transformation), but more flexible models are required to capture subtle local deformations. Such registration models [3, 32] may have large numbers of parameters, e.g., a large number of B-spline control points [30] or may even be non-parametric where vector fields are estimated [5, 22]. Spatial regularity can be achieved by appropriate constraints on displacement fields [14] or by parameterizing the transformation via integration of a sufficiently regular stationary or time-dependent velocity field [5, 15, 38, 8, 41]. Given sufficient regularization, diffeomorphic transformations can be assured in the continuum. A popular approach based on time-dependent velocity fields is LDDMM [5, 15].

Optimal LDDMM solutions are geodesics and minimize a geodesic distance. Consequentially, one may directly optimize over a geodesic's initial conditions in a shooting approach [40].

Most existing non-parametric image registration approaches use spatially-invariant regularizers. However, this may not be realistic. E.g., when registering inhale to exhale images of a lung one expects large deformations of the lung, but not of the surrounding tissue. Hence, a spatially-varying regularization would be more appropriate. As the regularizer encodes the deformation model this then allows anticipating different levels of deformation at different image locations.

While spatially-varying regularizers may be used in LDDMM variants [31] existing approaches do not allow for *time-varying* spatially-varying regularizers. However, such regularizers would be natural for large displacement as they can move with a deforming image. Hence, we propose a family of registration approaches with *spatio-temporal regularizers* based on advecting spatially-varying regularizers via an estimated spatio-temporal velocity field. Specifically, we extend LDDMM theory, where the original LDDMM model becomes a special case. In doing so, our entire model, including the spatio-temporal regularizer is expressed via the initial conditions of a partial differential equation. We propose three different approaches based on this model: 1) A model for which the regularizer is specified region-by-region. This would, for example, be natural when registering a labeled atlas image to a target image, as illustrated in Fig. 1. 2) A model in which the regularizer is estimated jointly with the spatio-temporal velocity fields. 3) A deep learning model which predicts the regularizer and the initial velocity field, thereby resulting in a very fast registration approach.

**Related Work** Only limited work [26, 27, 36, 39] on spatially-varying regularizers exists. The most related work [23] learns a spatially-varying regularizer for a stationary velocity field registration model. In [23] both the velocity field and the regularizer are assumed to be constant in time. In contrast, both are time-dependent in RDMM. Spatially-varying regularization has also been addressed from a Bayesian view in [34] by putting priors on B-spline transformation parameters. However, metric estimation in [34] is in a fixed atlas-space, whereas RDMM addresses general pairwise image registration.

**Contributions**: 1) We propose RDMM, a new registration model for large diffeomorphic deformations with a spatio-temporal regularizer capable of following deforming objects and thereby providing a more natural representation of deformations than existing non-parametric models, such as LDDMM. 2) Via a variational formulation we derive shooting equations that allow specifying RDMM solutions entirely based on their initial conditions: an initial momentum field and an initial spatially-varying regularizer. 3) We prove that diffeomorphisms can be obtained for RDMM in the continuum for sufficiently regular regularizers. 4) We explore an entire new family of registration models based on RDMM and provide optimization-based *and* very fast deep-learning-based approaches to estimate the initial conditions of these registration models. 5) We demonstrate the utility of our approach via experiments on synthetic data and on two 3D medical image datasets.

## 2  Standard LDDMM Model

LDDMM [5] is a non-parametric registration approach based on principles from fluid mechanics. It is based on the estimation of a spatio-temporal velocity field $v(t, x)$ from which the sought-for spatial transformation $\varphi$ can be computed via integration of $\partial_t \varphi(t, x) = v(t, \varphi(t, x))$. For appropriately regularized velocity fields [13], diffeomorphic transformations can be guaranteed. The optimization problem underlying LDDMM for images can be written as ($\nabla$ being the gradient; $\langle \cdot, \cdot \rangle$ indicating the inner product)

$$v^* = \operatorname*{argmin}_{v} \frac{1}{2} \int_0^1 \|v(t)\|_L^2 \, \mathrm{d}t + \mathrm{Sim}(I(1), I_1), \quad \text{s.t.} \quad \partial_t I + \langle \nabla I, v \rangle = 0; I(0) = I_0 \ . \quad (2.1)$$

Here, the goal is to register the source image $I_0$ to the target image $I_1$ in unit time. $\mathrm{Sim}(A, B)$ is a similarity measure between images, often sum of squared differences, normalized cross correlation, or mutual information. Furthermore, we note that $I(1, y) = I_0 \circ \varphi^{-1}(1, y)$, where $\varphi^{-1}$ denotes the inverse of $\varphi$ in the target image space. The evolution of this map can be expressed as

$$\partial_t \varphi^{-1} + D\varphi^{-1} v = 0 \,, \quad (2.2)$$

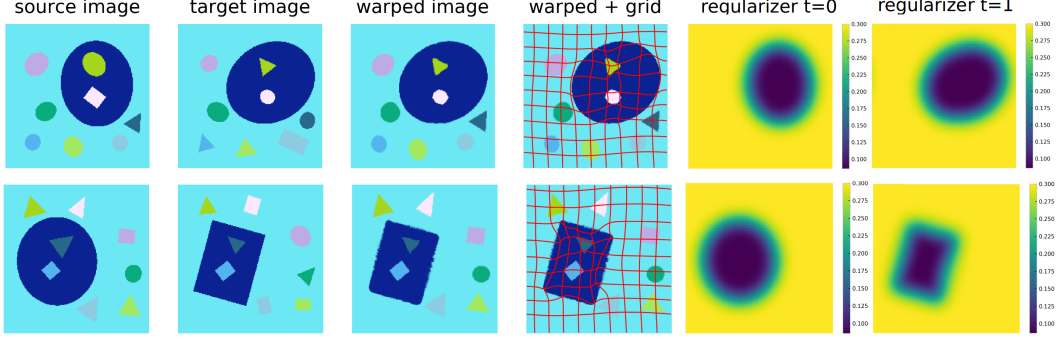

| source image | target image | warped image | warped + grid | regularizer t=0 | regularizer t=1 |

Figure 1: RDMM registration example. The goal is to register the dark blue area with high fidelity (i.e., allowing large local deformations), while assuring small deformations in the cyan area. Initially ($t = 0$), a spatially-varying regularizer (fifth column) is specified in the source image space, where dark blue indicates small regularization and yellow large regularization. Specifically, regularizer values indicate effective local standard deviations of a local multi-Gaussian regularizer. Since the transformation map and the regularizer are both advected according to the estimated velocity field, the shape of the regularizer follows the shape of the deforming dark blue region and is of the same shape as the region of interest in the target space at the final time ($t = 1$) (as can be seen in the second and the last columns). Furthermore, objects inside the dark blue region are indeed aligned well, whereas objects in the cyan region were not strongly deformed due to the larger regularization there.

where $D$ denotes the Jacobian. Equivalently, in Eq. (2.1), we directly advect the image [15, 40] via $\partial_t I + \langle \nabla I, v \rangle = 0$. To assure smooth transformations, LDDMM penalizes non-smooth velocity fields via the norm $\|v\|_L^2 = \langle Lv, v \rangle$, where $L$ is a differential operator.

At optimality of Eq. (2.1) the Euler-Lagrange equations are (div denoting the divergence)

$$\partial_t I + \langle \nabla I, v \rangle = 0, \ I(0) = I_0; \ \partial_t \lambda + \text{div}(\lambda v) = 0, \ \lambda(1) = \frac{\delta}{\delta I(1)} \text{Sim}(I(1), I_1) = 0; \ v = L^{-1}(\lambda \nabla I).$$
(2.3)

Here, $\lambda$ is the adjoint variable to $I$, also known as the scalar momentum [15, 40]. As $L^{-1}$ is a smoother it is often chosen as a convolution, i.e., $v = K \star (\lambda \nabla I)$. Note that $m(t, x) := \lambda \nabla I$ is the vector-valued momentum and thus $v = K \star m$. One can directly optimize over $v$ solving Eq. (2.3) [5] or regard Eq. (2.3) as a constraint defining a geodesic path [40] and optimize over all such solutions subject to a penalty on the initial scalar momentum as well as the similarity measure. Alternatively, one can express [1] these equations entirely with respect to the vector-valued momentum, $m$, resulting in the Euler-Poincaré equation for diffeomorphisms (EPDiff) [44]:

$$\partial_t m + \text{div}(v)m + Dv^T(m) + Dm(v) = 0, \ m(0) = m_0, v = K \star m,$$
(2.4)

which defines the evolution of the spatio-temporal velocity field based on the initial condition, $m_0$, of the momentum, from which the transformation $\varphi$ can be computed using Eq. (2.2). Both (2.3) and (2.4) can be used to implement shooting-based LDDMM [40, 35]. As LDDMM preserves the momentum, $\|v\|_L^2$ is constant over time and hence a *shooting formulation* can be written as

$$m(0)^* = \underset{m(0)}{\text{argmin}} \ \frac{1}{2} \|v(0)\|_L^2 + \text{Sim}(I(1), I_1),$$
(2.5)

subject to the EPDiff equation (2.4) including the advection of $\varphi^{-1}$, where $I(1) = I_0 \circ \varphi^{-1}(1)$.

A shooting-formulation has multiple benefits: 1) it allows for a compact representation of $\varphi$ via its initial conditions; 2) as the optimization is w.r.t. the initial conditions, a solution is a geodesic by construction; 3) these initial conditions can be predicted via deep-learning, resulting in very fast registration algorithms which inherit the theoretical properties of LDDMM [43, 42]. We therefore use this formulation as the starting point for RDMM in Sec 3.

# 3 Region-Specific Diffeomorphic Metric Mapping (RDMM)

In standard LDDMM approaches, the regularizer $L$ (equivalently the kernel $K$) is spatially invariant. While recent work introduced spatially-varying metrics in LDDMM, for stationary velocity fields, or for displacement-regularized registration [27, 31, 23, 26], all of these approaches use a temporally fixed regularizer. Hereafter, we generalize LDDMM by *advecting* a spatially-varying regularizer via the estimated spatio-temporal velocity field. Standard LDDMM is a special case of our model.

Following [23], we introduce $(V_i)_{i=0,\ldots,N-1}$ a finite family of reproducing kernel Hilbert spaces (RKHS) which are defined by the pre-defined Gaussian kernels $K_{\sigma_i}$ with $\sigma_0 < \ldots < \sigma_{N-1}$. We use a partition of unity $w_i(x,t), i = 0, \ldots, N-1$, on the image domain. As we want the kernel $K_{\sigma_i}$ to be active on the region determined by $w_i$ we introduce the vector field $v = \sum_{i=0}^{N-1} w_i \nu_i$ for $\nu_i \in V_i$. On this new space of vector fields, there exists a natural RKHS structure defined by

$$\|v\|_L^2 := \inf \left\{ \sum_{i=0}^{N-1} \|\nu_i\|_{V_i}^2 \mid v = \sum_{i=0}^{N-1} w_i \nu_i \right\}, \tag{3.1}$$

whose kernel is $K = \sum_{i=0}^{N-1} w_i K_{\sigma_i} w_i$. Thus the velocity reads (see suppl. 7.1 for the derivation)

$$v = K \star m \stackrel{\text{def.}}{=} \sum_{i=0}^{N-1} w_i K_{\sigma_i} \star (w_i m), w_i \geq 0, \tag{3.2}$$

which can capture multi-scale aspects of deformation [28]. In LDDMM, the weights are constant and pre-defined. Here, we allow spatially-dependent weights $w_i(x)$. In particular (see formulation below), we advect them via, $v(t, x)$ thereby making them spatio-temporal, i.e., $w_i(t, x)$. In this setup, weights only need to be specified at initial time $t = 0$. As the Gaussian kernels are fixed convolutions can still be efficiently computed in the Fourier domain.

We prove (see suppl. 7.4 for the proof) that, for sufficiently smooth weights $w_i$, the velocity field is bounded and its flow is a diffeomorphism. Following [23], to assure the smoothness of the initial weights we instead optimize over *initial pre-weights*, $h_i(0, x) \geq 0$, s.t. $w_i(0, x) = G_\sigma \star h_i(0, x)$, where $G_\sigma$ is a fixed Gaussian with a small standard deviation, $\sigma$. In addition, we constrain $\sum_{i=0}^{N-1} h_i^2(0, x)$ to locally sum to one. The optimization problem for our RDMM model then becomes

$$v^*, \{h_i^*\} = \operatorname*{argmin}_{v, \{h_i\}} \frac{1}{2} \int_0^1 \|v(t)\|_L^2 \, \mathrm{d}t + \mathrm{Sim}(I(1), I_1) + \mathrm{Reg}(\{h_i(0)\}), \tag{3.3}$$

subject to the constraints

$$\partial_t I + \langle \nabla I, v \rangle = 0, \ I(0) = I_0; \ \partial_t h_i + \langle \nabla h_i, v \rangle = 0, \ h_i(0) = (h_i)_0; \tag{3.4}$$

$$\nu_i = K_{\sigma_i} \star (w_i m); v = \sum_{i=0}^{N-1} w_i \nu_i; \ w_i = G_\sigma \star h_i.$$

As for LDDMM, we can compute the optimality conditions for Eq. (3.3) which we use for shooting.

**Theorem 1** (Image-based RDMM optimality conditions)**.** *With the adjoints $\gamma_i$ (for $h_i$) and $\lambda$ (for $I$) and the momentum $m := \lambda \nabla I + \sum_{i=0}^{N-1} \gamma_i \nabla h_i$ the optimality conditions for* (3.3) *are:*

$$\partial_t I + \langle \nabla I, v \rangle = 0, \ I(0) = I_0; \ \partial_t \lambda + \mathrm{div}(\lambda v) = 0, \ -\lambda(1) + \frac{\delta}{\delta I(1)} \mathrm{Sim}(I(1), I_1) = 0; \tag{3.5}$$

$$\partial_t h_i + \langle \nabla h_i, v \rangle = 0, \ h_i(0) = (h_i)_0; \tag{3.6}$$

$$\partial_t \gamma_i + \mathrm{div}(\gamma_i v) = G_\sigma \star (m \cdot \nu_i), \ \gamma_i(0) + \frac{\delta}{\delta h_i(0)} \mathrm{Reg}(\{h_i(0)\}) = 0. \tag{3.7}$$

*subject to*

$$\nu_i = K_{\sigma_i} \star (w_i m); \ v = \sum_{i=0}^{N-1} w_i \nu_i; \ w_i = G_\sigma \star h_i. \tag{3.8}$$

**Theorem 2** (Momentum-based RDMM optimality conditions)**.** *The RDMM optimality conditions of Thm.* (1) *can be written entirely w.r.t. the momentum (as defined in Thm* (1)*). They are:*

$$\partial_t \varphi^{-1} + D\varphi^{-1} v = 0, \ \varphi^{-1}(0, x) = x, \tag{3.9}$$

$$\partial_t m + \text{div}(v)m + Dv^T(m) + Dm(v) = \sum_{i=0}^{N-1} G_\sigma \star (m \cdot \nu_i) \nabla h_i, \ m(0) = m_0, \tag{3.10}$$

*where* $h_i(t, x) = h_i(0, x) \circ \varphi(t, x)^{-1}$ *and subject to the constraints of Eq.* (3.8) *which define the relationship between the velocity and the momentum.*

For spatially constant pre-weights, we recover EPDiff from the momentum-based RDMM optimality conditions. Instead of advecting $h_i$ via $\varphi(t, x)^{-1}$, we can alternatively advect the pre-weights directly, as $\partial_t h_i + \langle \nabla h_i, v \rangle = 0, \ h_i(0) = (h_i)_0$. For the image-based and the momentum-based formulations, the velocity field $v$ is obtained by smoothing the momentum, $v = \sum_{i=0}^{N-1} w_i K_{\sigma_i} \star (w_i m)$.

### Regularization of the Regularizer

Given the standard deviations $\sigma_0 < \ldots < \sigma_{N-1}$ of Eq. (3.2), assigning larger weights to the Gaussians with larger standard deviation will result in smoother (i.e., more regular) and therefore simpler transformations. To encourage choosing simpler transformations we follow [23], where a simple optimal mass transport (OMT) penalty on the weights is used. Such an OMT penalty is sensible as the non-negative weights, $\{w_i\}$, sum to one and can therefore be considered a discrete probability distribution. The chosen OMT penalty is designed to measure differences from the probability distribution assigning all weight to the Gaussian with the largest standard deviation, i.e., $w_{N-1} = 1$ and all other weights being zero. Specifically, the OMT penalty of [23] is $\left| \log \frac{\sigma_{N-1}}{\sigma_0} \right|^{-s} \sum_{i=0}^{N-1} w_i \left| \log \frac{\sigma_{N-1}}{\sigma_i} \right|^s$, where $s$ is a chosen power. To make the penalty more consistent with penalizing weights for a standard multi-Gaussian regularizer (as our regularizer contains effectively the weights squared) we do not penalize the weights directly, but instead penalize their squares using the same form of OMT penalty. Further, as the regularization only affects the initial conditions for the pre-weights, the evolution equations for the optimality conditions (i.e., the modified EPDiff equation) do not change. Additional regularizers, such as total variation terms as proposed in [23], are possible and easy to integrate into our RDMM framework as they only affect initial conditions. For simplicity, we focus on regularizing via OMT.

### Shooting Formulation

As energy is conserved (see suppl. 7.3 for the proof) the momentum-based shooting formulation becomes

$$m(0)^*, \{h_i(0)^*\} = \underset{m(0), \{h_i(0)\}}{\text{argmin}} \ \frac{1}{2} \|v(0)\|_L^2 + \text{Sim}(I(1), I_1) + \text{Reg}(\{h_i(0)\}), \tag{3.11}$$

subject to the evolution equations of Thm. 2. Similarly, the shooting formulation can use the image-based evolution equations of Thm. 1 where optimization would be over $\lambda(0)$ instead of $m(0)$.

## 4 Learning Framework

The parameters for RDMM, i.e., the initial momentum and the initial pre-weights, can be obtained by numerical optimization, either over the momentum *and* the pre-weights or only over the momentum if the pre-weights are prescribed. Just as for the LDDMM model, such a numerical optimization is computationally costly. Consequentially, various deep-learning (DL) approaches have been proposed to instead predict displacements [4, 6], stationary velocity [29] or momentum fields [43, 23]. Supervised [43, 42] and unsupervised [16, 11, 18, 4, 10] DL registration approaches exist. All of them are fast as only a regression solution needs to be evaluated at test time and no further numerical optimization is necessary. Additionally, such DL models benefit from learning over an entire population instead of relying only on information from given image-pairs.

Most non-parametric registration approaches are not invariant to affine transformations based on the chosen regularizers. Hence, for such non-parametric methods, affine registration is typically performed first as a pre-registration step to account for large, global displacements or rotations. Similarly, we make use of a two-step learning framework (Fig. 2 (left)) learning affine transformations

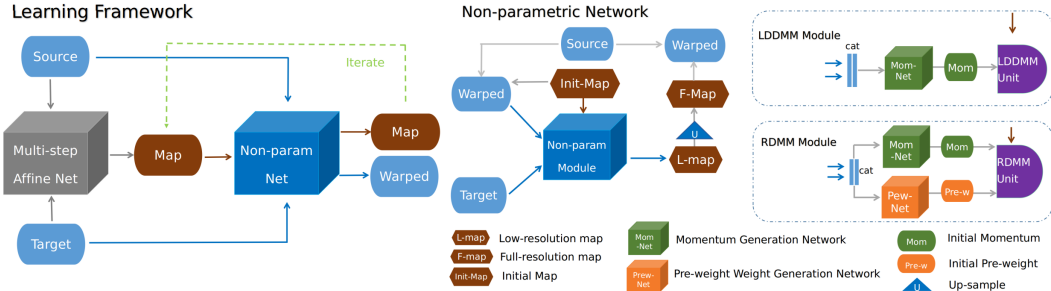

Figure 2: Illustration of the learning framework (left) and its non-parametric registration component (right). A multi-step affine-network first predicts the affine transformation map [33] followed by an iterable non-parametric registration to estimate the final transformation map. The LDDMM component uses one network to generate the initial momentum. RDMM also uses a second network to predict the initial regularizer pre-weights. We integrate the RDMM evolution equations at low-resolution (based on the predicted initial conditions) to save memory. The final transformation map is obtained via upsampling. See suppl. 7.6 for the detailed structure of the LDDMM/RDMM units.

and subsequent non-parametric deformations separately. For the affine part, a multi-step affine network is used to predict the affine transformation following [33]. For the non-parametric part, we use two different deep learning approaches, illustrated in the right part of Fig. 2, to predict 1) the LDDMM initial momentum, $m_0$, in Eq. (2.5) and 2) the RDMM initial momentum, $m_0$, and the pre-weights, $\{h_i(0)^*\}$, in Eq. (3.11). Overall, including the affine part, we use two networks for LDDMM prediction and three networks for RDMM prediction.

We use low-resolution maps and map compositions for the momentum and the pre-weight networks. This reduces computational cost significantly. The final transformation map is obtained via up-sampling, which is reasonable as we assume smooth transformations. We use 3D UNets [9] for momentum and pre-weight prediction. Both the affine and the non-parametric networks can be iterated to refine the prediction results: i.e. the input source image and the initial map are replaced with the currently warped image and the transformation map respectively for the next iteration.

During training of the non-parametric part, the gradient is first backpropagated through the differentiable interpolation operator, then through the LDDMM/RDMM unit, followed by the momentum generation network and the pre-weight network.

*Inverse Consistency*: For the DL approaches we follow [33] and compute bidirectional (source to target denoted as $^{st}$ and target to source denoted as $^{ts}$) registration losses and an additional symmetry loss, $\|(\varphi^{st})^{-1} \circ (\varphi^{ts})^{-1} - id\|_2^2$, where $id$ refers to the identity map. This encourages symmetric consistency.

## 5   Experimental Results and Setup

**Datasets**: To demonstrate the behavior of RDMM, we evaluate the model on three datasets: 1) a synthetic dataset for illustration, 2) a 3D computed tomography dataset (CT) of a lung, and 3) a large 3D magnetic resonance imaging (MRI) dataset of the knee from the Osteoarthritis Initiative (OAI). *The synthetic dataset* consists of three types of shapes (rectangles, triangles, and ellipses). There is one foreground object in each image with two objects inside and at most five objects outside. Each source image object has a counterpart in the target image; the shift, scale, and rotations are random. We generated 40 image pairs of size $200^2$ for evaluation. Fig. 1 shows example synthetic images. *The lung dataset* consists of 49 inspiration/expiration image pairs with lung segmentations. Each image is of size $160^3$. We register from the expiration phase to the inspiration phase for all 49 pairs. *The OAI dataset* consists of 176 manually labeled MRI from 88 patients (2 longitudinal scans per patient) and 22,950 unlabeled MR images from 2,444 patients. Labels are available for femoral and tibial cartilage. We divide the patients into training (2,800 pairs), validation (50 pairs) and testing groups (300 pairs), with the same evaluation settings as for the cross-subject experiments in [33].

**Deformation models**: Affine registration is performed before each LDDMM/RDMM registration.

*Affine model*: We implemented a multi-scale affine model solved via numerical optimization and a multi-step deep neural network to predict the affine transformation parameters.

*Family of non-parametric models*: We implemented both *optimization* and *deep-learning* versions of a family of non-parametric registration methods: a vector-momentum based stationary velocity field model (vSVF) ($v(x)$, $w = const$), LDDMM ($v(t,x)$, $w = const$), and RDMM ($v(t,x)$, $w(t,x)$). We use the dopri5 solver using the adjoint sensitivity method [7] to integrate the evolution equations in time. For solutions based on numerical optimization, we use a multi-scale strategy with L-BGFS [19] as the optimizer. For the deep learning models, we compute solutions for a low-resolution map (factor of 0.5) which is then upsampled. We use Adam [17] for optimization.

**Image similarity measure**: We use multi-kernel Localized Normalized Cross Correlation (mk-LNCC) [33]. mk-LNCC computes localized normalized cross correlation (NCC) with different window sizes and combines these measures via a weighted sum.

**Weight visualization**: To illustrate the behavior of the RDMM model, we visualize the estimated standard deviations, i.e.the square root of the local variance $\sigma^2(x) = \sum_{i=0}^{N-1} w_i^2(x)\sigma_i^2$.

**Estimation approaches**: To illustrate different aspects of our approach we perform three types of RDMM registrations: 1) *registration with a pre-defined regularizer* (Sec. 5.1), 2) *registration with simultaneous optimization of the regularizer*, via optimization of the initial momentum and pre-weights (Sec. 5.2), and 3) *registration via deep learning predicting* the initial momentum and regularizer pre-weights (Sec. 5.3). Detailed settings for all approaches are in the suppl. 7.8.

## 5.1 Registration with a pre-defined regularizer

To illustrate the base capabilities of our models, we prescribe an initial spatially-varying regularizer in the source image space. We show experimental results for pair-wise registration of the synthetic data as well as for the 3D lung volumes.

Fig. 1 shows the registration result for an example synthetic image pair. We use small regularization in the blue area and large regularization in the surrounding area. As expected, most of the deformations occur inside the blue area as the regularizer is more permissive there. We also observe that the regularizer is indeed advected with the image. For the real lung image data we use a small regularizer inside the lung (as specified by the given lung mask) and a large regularizer in the surrounding tissue. Fig. 3 shows that most of the deformations are indeed inside the lung area while the deformation outside the lung is highly regularized as desired. We evaluate the Dice score between the warped lung and the target lung, achieving $95.22\%$ on average (over all inhalation/exhalation pairs). Fig. 3 also shows the determinant of the Jacobian of the transformation map $J_{\varphi^{-1}}(x) := |D\varphi^{-1}(x)|$ (defined in target space): the lung region shows small values (illustrating expansion) while other region are either volume preserved (close to 1) or compressed (bigger than 1). Overall the deformations are smooth.

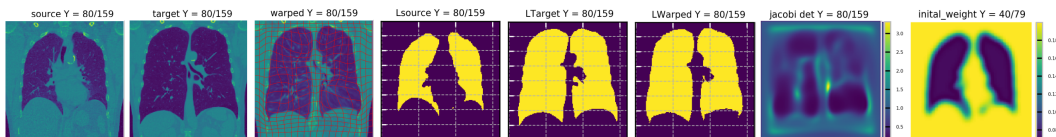

Figure 3: RDMM lung registration result with a pre-defined regularizer. Lung images at expiration (source) are registered to the corresponding inspiration (target) images. Last column: Inside the lung a regularizer with small standard deviation ($\sigma_i = \{0.04, 0.06, 0.08\}$, $h_0^2 = \{0.1, 0.4, 0.5\}$) and outside the lung with large standard deviation is used ($\sigma_i = \{0.2\}$, $h_0^2 = \{1.0\}$). Deformations are largely inside the lung, the surrounding tissue is well regularized as expected. Columns 1 to 3 show results in image space while columns 4 to 6 refer to the results in label space. The second to last column shows the determinant of the Jacobian of the spatial transformation, $\varphi^{-1}$.

## 5.2 Registration with an optimized regularizer

In contrast to the experiments in Sec. 5.1, we jointly estimate the initial momentum *and* the initial regularizer pre-weights for pairwise registration. We use the synthetic data and the knee MRIs.

Fig. 4 shows that the registration warped every object in the source image to the corresponding object in the target image. The visualization of the initial regularizer shows that low regularization is

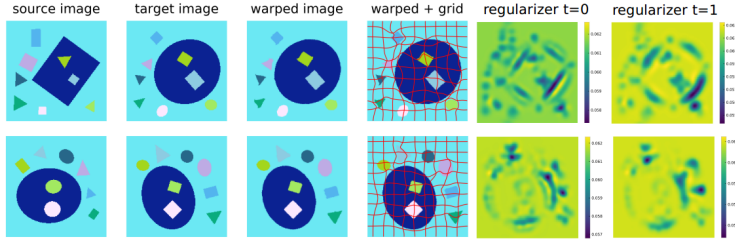

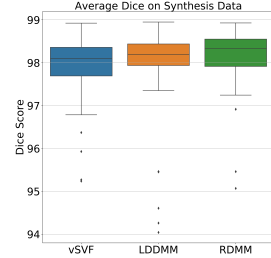

Figure 4: Illustration of the RDMM registration results with an optimized regularizer on the synthetic dataset. All objects are warped from the source image space to the target image space. The last two columns show the regularizer ($\sigma(x)$) at $t = 0$ and $t = 1$ respectively.

Figure 5: Average Dice scores for all objects. Left to right: SVF, LDDMM and RDMM.

| Method | OAI Dataset Dice | Folds | Time (s) |
|---|---|---|---|
| *Affine Methods* | | | |
| affine-NiftyReg | 30.43 (12.11) | 0 | 45 |
| affine-opt | 34.49 (18.07) | 0 | 8 |
| affine-net | **44.58** (7.74) | 0 | 0.20 |
| *Optimization Methods* | | | |
| Demons[38, 37] | 63.47 (9.52) | 0.56 | 114 |
| SyN[2, 1] | 65.71 (15.01) | 0 | 1330 |
| NiftyReg-NMI[25, 20, 30, 21] | 59.65 (7.62) | 0 | 143 |
| NiftyReg-LNCC | 67.92 (5.24) | 35.19 | 270 |
| vSVF-opt | 67.35 (9.73) | 0 | 79 |
| LDDMM-opt | 67.72 (8.94) | 0 | 457 |
| RDMM-opt | **68.18** (8.36) | 17.37 | 627 |
| *Learning-based Methods* | | | |
| VoxelMorph[10](with aff) | 66.08 (5.13) | 3.31 | 0.31 |
| vSVF-net [33] | 67.59 (4.47) | 0.39 | 0.62 |
| LDDMM-net | 67.63 (4.51) | 0 | 0.85 |
| RDMM-net | **67.94** (4.40) | 0.47 | 1.1 |

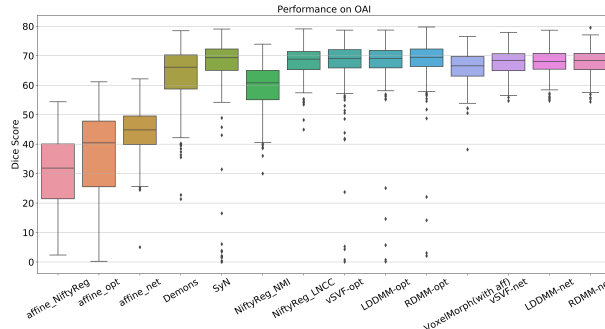

Figure 6: Comparison of registration methods for cross-subject registrations on the OAI dataset based on Dice scores. *-opt* and *-net* refer to optimization- and DL-based methods respectively. For all DL methods, we report performance after two-step refinement. *Folds* refers to the absolute value of the sum of the determinant of the Jacobian in the folding area (i.e., where the determinant of the Jacobian is negative); *Time* refers to the average registration time for a single image pair.

assigned close to object edges making them locally deformable. The visualizations of the regularizer at the initial time point ($t = 0$) and the final time point ($t = 1$) show that it deforms with the image. That low regularization values are localized is sensible as the OMT regularizer prefers spatially sparse solutions (in the sense of sparsely assigning low levels of regularity). If the desired deformation model is piecewise constant our RDMM model could readily be combined with a total variation penalty as in [23]. Fig. 5 compares average Dice scores for all objects for vSVF, LDDMM and RDMM separately. They all achieve high and comparable performance indicating good registration quality. But only RDMM provides additional information about local regularity.

We further evaluate RDMM on 300 images pairs from the OAI dataset. The *optimization methods* section of Tab. 6 compares registration performance for different optimization-based algorithms. RDMM achieves high performance. While RDMM is in theory diffeomorphic, we observe some foldings, whereas no such foldings appear for LDDMM and SVF. This is likely due to inaccuracies when discretizing the evolution equations and when discretizing the determinant of the Jacobian. Further, RDMM may locally exhibit stronger deformations than LDDMM or vSVF, especially when the local regularization (via OMT) is small, making it numerically more challenging. Most of the folds appear at the image boundary (and are hence likely due to boundary discretization artifacts) or due to anatomical inconsistency in the source and target images, where large deformations may be estimated.

## 5.3 Registration with a learnt regularizer via deep learning

Finally, we evaluate our learning framework for non-parametric registration approaches on the OAI dataset. For vSVF, we follow [33] to predict the momentum. We implement the same approach for LDDMM, where the vSVF inference unit is replaced by one for LDDMM (i.e., instead of advecting via a stationary velocity field we integrate EPDiff). Similarly, for RDMM, the inference unit is replaced by the evolution equation for RDMM and we use an additional network to predict the regularizer pre-weights. Fig. 6 shows the results. The non-parametric DL models achieve comparable

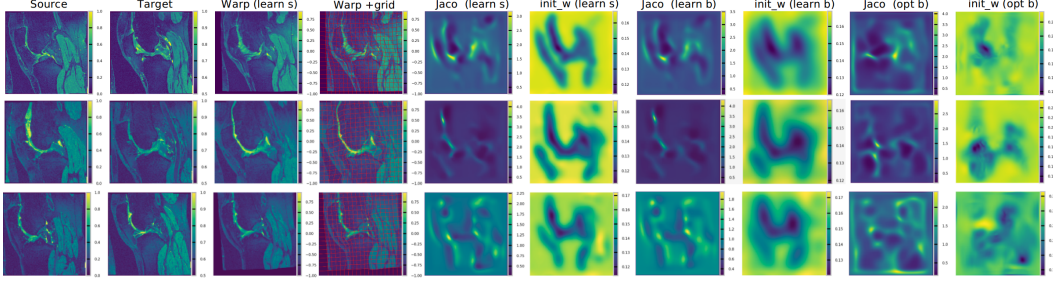

Figure 7: Illustration of RDMM registration results on the OAI dataset. "* s" and "* b" refer to the regularizer with $\sigma$ in $G_\sigma$ set to $0.04$ and $0.06$ respectively; "learn *" and "opt *" refer to a learnt regularizer and an optimized one respectively; "Jaco" refers to the absolute value of the determinant of the Jacobian; "init_w" refers to the initial weight map of the regularizer (as visualized via $\sigma(x)$). The first four columns refer to registration results in image space.

performance to their optimization counterparts, but are much faster, while learning-based RDMM simultaneously predicts the initial regularizer which captures aspects of the knee anatomy. Fig. 7 shows the determinant of the Jacobian of the transformation map. It is overall smooth and folds are much less frequent than for the corresponding optimization approach, because the DL model penalizes transformation asymmetry. Fig. 7 also clearly illustrates the benefit of training the DL model based on a large image dataset: compared with the optimization approach (which only works on individual image pairs), the initial regularizer predicted by the deep network captures anatomically meaningful information much better: the bone (femur and tibia) and the surrounding tissue show large regularity.

# 6   Conclusion and Future Work

We introduced RDMM, a generalization of LDDMM registration which allows for spatially-varying regularizers advected with a deforming image. In RDMM, both the estimated velocity field and the estimated regularizer are time- and spatially-varying. We used a variational approach to derive shooting equations which generalize EPDiff and allow the parameterization of RDMM using only the initial momentum and regularizer pre-weights. We also prove that diffeomorphic transformation can be obtained for RDMM with sufficiently regular regularizers. Experiments with pre-defined, optimized, and learnt regularizers show that RDMM is flexible and its solutions can be estimated quickly via deep learning.

Future work could focus on numerical aspects and explore different initial constraints, such as total-variation constraints, depending on the desired deformation model. Indeed, a promising avenue of research consists in learning regularizers which include more physical/mechanical a-priori information in the deformation model. For instance, a possible first step in this direction consists in parameterizing non-isotropic kernels to favor deformations in particular directions.

**Acknowledgements** Research reported in this work was supported by the National Institutes of Health (NIH) and the National Science Foundation (NSF) under award numbers NSF EECS-1711776 and NIH 1R01AR072013. The content is solely the responsibility of the authors and does not necessarily represent the official views of the NIH or the NSF. We would also like to thank Dr. Raúl San José Estépar for providing the lung data.

## Footnotes

[1]Thm. (2) in suppl. 7.2 provides a more generalized derivation.

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
