[Supplementary Material · rdmm_suppl.pdf]

# 7    Supplementary Material

This supplementary material provides additional details illustrating the proposed approach. We start by deriving the form of the smoothing kernel for the RDMM model in Sec. 7.1. Based on this kernel form we can then detail the derivation of the RDMM optimality conditions in Sec. 7.2. In Sec. 7.3, we prove that the regularization energy is conserved over time, which allows formulating our RDMM shooting strategy based on initial conditions only. Sec. 7.4 details the good theoretical behavior of our model. Sec. 7.5 describes the optimization/training strategy with regard to the the initial pre-weight regularization. Sec. 7.6 visualizes the inference process of the LDDMM/RDMM method. Sec. 7.7 analyzes the behavior of the OMT term. Lastly, Sec. 7.8 details the settings of our experiments.

## 7.1    Variational Derivation of the Smoothing Kernel

The derivation of our RDMM model makes use of a smoothing kernel of the form $K = \sum_{i=0}^{N-1} w_i K_{\sigma_i} w_i$. This kernel form is a direct consequence of the definition of our variational definition of the smoothing kernel.

Recall that similar to [23] we define

$$\|v\|_L^2 := \inf \left\{ \sum_{i=0}^{N-1} \|\nu_i\|_{V_i}^2 \mid v = \sum_{i=0}^{N-1} w_i \nu_i \right\}, \tag{7.1}$$

for a given velocity field $v$. To compute an explicit form of the norm $\|v\|_L^2$ we need to solve the constrained optimization problem of this definition. Specifically, we introduce the vector-valued Lagrange multiplier $m$. Thus the Lagrangian, $\mathcal{L}$, becomes[2]

$$\mathcal{L}(\{\nu_i\}, m) = \sum_{i=0}^{N-1} \frac{1}{2} \|\nu_i\|_{V_i}^2 - \langle m, w_i \nu_i - v \rangle$$

$$= \sum_{i=0}^{N-1} \frac{1}{2} \langle L_i \nu_i, \nu_i \rangle - \langle m, w_i \nu_i - v \rangle. \tag{7.2}$$

The variation of the Lagrangian is

$$\delta\mathcal{L}(\{\nu_i\}, m; \{\delta\nu_i\}, \delta m) = \sum_{i=0}^{N-1} \langle L_i \nu_i, \delta\nu_i \rangle - \langle \delta m, w_i \nu_i - v \rangle - \langle m, w_i \delta\nu_i \rangle. \tag{7.3}$$

By collecting all the terms, the optimality conditions (i.e., where the variation vanishes) are

$$L_i \nu_i = w_i m, \ \forall i \quad \text{and} \quad v = \sum_{i=0}^{N-1} w_i \nu_i. \tag{7.4}$$

Hence, we can write the norm $\|v(t)\|_L^2$ in the following form :

$$\|v\|_L^2 = \sum_{i=0}^{N-1} \langle L_i \nu_i^*, \nu_i^* \rangle = \sum_{i=0}^{N-1} \langle w_i m, L_i^{-1} w_i m \rangle = \sum_{i=0}^{N-1} \langle m, w_i K_{\sigma_i} \star (w_i m) \rangle. \tag{7.5}$$

Consequentially, the associated kernel is $K = \sum_{i=0}^{N-1} w_i K_{\sigma_i} w_i$. Assuming the kernel can be written as a convolution, we can therefore express the velocity as:

$$v = \sum_{i=0}^{N-1} w_i \nu_i = \sum_{i=0}^{N-1} w_i K_{\sigma_i} \star (w_i m). \tag{7.6}$$

## 7.2    Optimality Conditions

In this section we derive the RDMM optimality conditions. Both for the image-based and the momentum-based cases. Recall that the overall registration energy of RDMM can be written as:

$$E(v, I, \{h_i\}) = \frac{1}{2} \int_0^1 \|v(t)\|_L^2 \, \mathrm{d}t + \mathrm{Sim}(I(1), I_1) + \mathrm{Reg}(\{h_i(0)\}) \tag{7.7}$$

under the constraints[3]

$$\begin{cases} I_t + \langle \nabla I, v \rangle = 0, \ I(0) = I_0, \\ (h_i)_t + \langle \nabla h_i, v \rangle = 0, h_i(0) = (h_i)_0, \\ w_i = G_\sigma \star h_i, \\ \nu_i = K_{\sigma_i} \star (w_i m), \\ v = \sum_{i=0}^{N-1} w_i \nu_i . \end{cases} \tag{7.8}$$

**Proof of Thm.** (1)

We compute the variations of the Lagrangian, $\mathcal{L}$ to the energy (i.e., where constraints are added via Lagrangian multipliers) with respect to $v$, $\lambda$, $I$, $\{h_i\}$ and $\{\gamma_i\}$:

$$\begin{aligned} \delta \mathcal{L} =& \frac{\partial}{\partial \epsilon} \mathcal{L} \left( v + \epsilon dv, I + \epsilon dI, \{h_i + \epsilon dh_i\}, \lambda + \epsilon d\lambda, \{\gamma_i + \epsilon d\gamma_i)\} \right)|_{\epsilon=0} \\ =& \int_0^1 \frac{1}{2} \delta(\|v(t)\|_L^2) - \langle d\lambda, I_t + (DI)v \rangle - \langle \lambda, dI_t + (DdI)v + (DI)dv \rangle \\ & - \sum_{i=0}^{N-1} \{ \langle d\gamma_i, h_{it} + (Dh_i)v \rangle + \langle \gamma_i, dh_{it} + (Ddh_i)v + (Dh_i)dv \rangle \} \ \mathrm{dt} \\ & + \left\langle \frac{\delta}{\delta I(1)} \operatorname{Sim}(I(1), I_1), dI(1) \right\rangle + \sum_{i=0}^{N-1} \left\langle \frac{\delta}{\delta h_i(0)} \operatorname{Reg}(\{h_i(0)\}), dh_i(0) \right\rangle . \end{aligned} \tag{7.9}$$

We use

$$\int_0^1 \langle \lambda, dI_t \rangle \, dt = \int_0^1 \langle -\lambda_t, dI \rangle \, dt + \langle \lambda, dI \rangle_0^1 . \tag{7.10}$$

According to Green's theorem and assuming $v$ vanishes on the boundary, we get

$$\langle \lambda, (DdI)v \rangle = \langle -div(\lambda v), dI \rangle + \int_{\partial\Omega} dI \lambda v \cdot dS = \langle -div(\lambda v), dI \rangle . \tag{7.11}$$

Similarly, we have

$$\int_0^1 \langle \gamma_i, dh_{it} \rangle \, dt = \int_0^1 \langle -\gamma_{it}, dh_i \rangle \, dt + \langle \gamma_i, dh_i \rangle_0^1 \tag{7.12}$$

$$\langle \gamma_i, (Ddh_i)v \rangle = \langle -div(\gamma_i v), dh_i \rangle + \int_{\partial\Omega} dh_i \gamma_i v \cdot dS = \langle -div(\gamma_i v), dh_i \rangle . \tag{7.13}$$

Now, Eq. (7.9) reads

$$\begin{aligned} \delta E =& \int_0^1 \frac{1}{2} \delta(\|v(t)\|_L^2) - \langle d\lambda, I_t + (DI)v \rangle + \langle \lambda_t + div(\lambda v), dI \rangle \\ & + \sum_{i=0}^{N-1} \{ -\langle d\gamma_i, h_{it} + (Dh_i)v \rangle + \langle \gamma_{it} + div(\gamma_i v), dh_i \rangle \} - \left\langle \lambda \nabla I + \sum_{i=0}^{N-1} \gamma_i \nabla h_i, dv \right\rangle \ \mathrm{dt} \\ & - \langle \lambda, dI \rangle_0^1 - \sum_{i=0}^{N-1} \langle \gamma_i, dh_i \rangle_0^1 \\ & + \left\langle \frac{\delta}{\delta I(1)} \operatorname{Sim}(I(1), I_1), dI(1) \right\rangle + \sum_{i=0}^{N-1} \left\langle \frac{\delta}{\delta h_i(0)} \operatorname{Reg}(\{h_i(0)\}), dh_i(0) \right\rangle . \end{aligned} \tag{7.14}$$

We first collect $dI(1)$ and $dh_i(0)$ to obtain the final condition on $\lambda$ and the initial condition on $\gamma$:

$$\begin{cases} -\lambda(1) + \frac{\delta}{\delta I(1)} \operatorname{Sim}(I(1), I_1) = 0, \\ \gamma_i(0) + \frac{\delta}{\delta h_i(0)} \operatorname{Reg}(\{h_i(0)\}) = 0 . \end{cases} \tag{7.15}$$

Next, we work on $\int_0^1 \frac{1}{2}\delta(\|v(t)\|_L^2)\,dt$. Remember, we have $v = K \star m \overset{\text{def.}}{=} \sum_{i=0}^{N-1} w_i \nu_i$, where $\nu_i = K_{\sigma_i} \star (w_i m), \quad w_i \geq 0$, thus

$$\int_0^1 \frac{1}{2}\delta(\|v(t)\|_L^2)\,dt = \int_0^1 \frac{1}{2}\langle dm, v\rangle + \frac{1}{2}\langle m, \sum_{i=0}^{N-1} w_i d\nu_i + \nu_i dw_i\rangle\,dt. \qquad (7.16)$$

Note that for radially symmetric kernels (such as Gaussian kernels) $K = \overline{K}$, $\langle K \ast a, b\rangle = \langle \overline{K} \ast b\rangle$

$$\begin{aligned}
\langle K \ast a, b\rangle &= \int_{x=-\infty}^{\infty} \left(\int_{y=-\infty}^{\infty} K(x-y)a(y)\right) b(x)dx \\
&= \int_{y=-\infty}^{\infty} a(y) \int_{x=-\infty}^{\infty} K(x-y)b(x)dxdy \\
&= \int_{y=-\infty}^{\infty} a(y) \int_{x=-\infty}^{\infty} \underbrace{\overline{K}(y-x)}_{\overline{K}(x):K(-x)} b(x)dxdy \qquad (7.17) \\
&= \int_{x=-\infty}^{\infty} a(x) \int_{y=-\infty}^{\infty} \overline{K}(x-y)b(y)dydx \\
&= \langle a, \overline{K} \ast b\rangle.
\end{aligned}$$

Thus, we can get

$$\begin{aligned}
\frac{1}{2}\langle \nu_i dw_i + w_i d\nu_i, m\rangle &= \frac{1}{2}\langle dw_i K_{\sigma_i} \star (w_i m) + w_i K_{\sigma_i} \star (dw_i m + w_i dm), m\rangle \\
&= \frac{1}{2}\langle m^T K_{\sigma_i} \star (w_i m), dw_i\rangle + \frac{1}{2}\langle w_i m, K_{\sigma_i} \star (dw_i m)\rangle + \frac{1}{2}\langle w_i m, K_{\sigma_i} \star (w_i dm)\rangle \\
&= \frac{1}{2}\langle m^T K_{\sigma_i} \star (w_i m), dw_i\rangle + \frac{1}{2}\langle m^T K_{\sigma_i} \star (w_i m), dw_i\rangle + \frac{1}{2}\langle w_i K_{\sigma_i} \star (w_i m), dm\rangle \\
&= \langle G_\sigma \star (m^T \nu_i), dh_i\rangle + \frac{1}{2}\langle w_i \nu_i, dm\rangle.
\end{aligned}$$
$$(7.18)$$

Substituting Eq. (7.18) into Eq. (7.16), we get

$$\int_0^1 \frac{1}{2}\delta(\|v(t)\|_L^2)\,dt = \int_0^1 \langle dm, v\rangle + \sum_{i=0}^{N-1} \langle G_\sigma \star (m^T \nu_i), dh_i\rangle\,dt. \qquad (7.19)$$

Next, we decompose the $\langle \lambda \nabla I + \sum_{i=0}^{N-1} \gamma_i \nabla h_i, dv\rangle$ terms. We define the momentum, $m = \lambda \nabla I + \sum_{i=0}^{N-1} \gamma_i \nabla h_i$.

$$\begin{aligned}
\langle \lambda \nabla I + \sum_{i=0}^{N-1} \gamma_i \nabla h_i, dv\rangle &= \langle m, \sum_{i=0}^{N-1} dw_i K_{\sigma_i} \star (w_i m) + w_i K_{\sigma_i} \star (dw_i m + w_i dm)\rangle \\
&= \sum_{i=0}^{N-1} \langle m^T K_{\sigma_i} \star (w_i m), dw_i\rangle + \langle m^T K_{\sigma_i} \ast (w_i m), dw_i\rangle + \langle w_i K_{\sigma_i} \star (w_i m), dm\rangle \\
&= \sum_{i=0}^{N-1} \langle G_\sigma \star [m^T K_{\sigma_i} \star (w_i m) + m^T K_{\sigma_i} \star (w_i m)], dh_i\rangle + \langle w_i K_{\sigma_i} \star (w_i m), dm\rangle \\
&= \sum_{i=0}^{N-1} 2\langle G_\sigma \star (m^T \nu_i), dh_i\rangle + \langle w_i \nu_i, dm\rangle.
\end{aligned}$$

Now, we can collect the variation $dh_i$ for $h_i$ and $dm$ for $m$ and obtain the optimality conditions

$$-G_\sigma \star (m^T \nu_i) + \gamma_{it} + div(\gamma_i v) = 0, \qquad (7.20)$$

$$v - \sum_{i=0}^{N-1} w_i \nu_i = 0. \qquad (7.21)$$

Finally, we get the optimality conditions for image-based RDMM derived from Eq. (7.7) and Eq. (7.8):

$$
\begin{cases}
I_t + \langle \nabla I, v \rangle = 0, \ I(0) = I_0 \,, \\
h_{it} + \langle \nabla h_i, v \rangle = 0, \ h_i(0) = (h_i)_0 \,, \\
\lambda_t + \operatorname{div}(\lambda v) = 0 \,, \\
\gamma_{it} + \operatorname{div}(\gamma_i v) = G_\sigma \star (m \cdot \nu_i) \,, \\
-\lambda(1) + \frac{\delta}{\delta I(1)} \operatorname{Sim}(I(1), I_1) = 0 \,, \\
\gamma_i(0) + \frac{\delta}{\delta h_i(0)} \operatorname{Reg}(\{h_i(0)\}) = 0 \,,
\end{cases}
\tag{7.22}
$$

where $\nu_i = K_{\sigma_i} \star (w_i m)$ and $m = \lambda \nabla I + \sum_{i=0}^{N-1} \gamma_i \nabla h_i$.

**Proof of Thm. (2)** We now derive the optimality conditions for the momentum-based formulation of RDMM. We start by taking the time derivative of the momentum and obtain

$$
-m_t \quad = \quad -(\lambda \nabla I)_t - \Big( \sum_{i=0}^{N-1} \nabla h_i \gamma_i \Big)_t
\tag{7.23}
$$

$$
= \quad -\lambda_t \nabla I - \lambda \nabla I_t - \sum_{i=0}^{N-1} \{ \gamma_{it} \nabla h_i + \gamma_i \nabla(h_{it}) \} \,.
\tag{7.24}
$$

By substituting the time derivatives $\lambda_t$, $I_t$, $\gamma_{it}$, and $h_{it}$ from Eq. (7.22) we obtain

$$
-m_t = \operatorname{div}(\lambda v) \nabla I + \lambda \nabla(\nabla I^T v) + \sum_{i=0}^{N-1} \Big[ \operatorname{div}(\gamma_i v) - G_\sigma \star (m^T \nu_i) \Big] \nabla h_i + \gamma_i \nabla(\nabla h_i^T v) \,.
\tag{7.25}
$$

Using the following two relations,

$$
\operatorname{div}(\lambda v) = \nabla \lambda^T v + \lambda \operatorname{div}(v) \quad \text{and} \quad \nabla(\nabla I^T v) = H I v + (Dv)^T \nabla I
\tag{7.26}
$$

where $D$ denotes the Jacobian and $H$ the Hessian, we can rewrite Eq. (7.25) as

$$
-m_t \quad = \quad ((\nabla \lambda)^T v + \lambda \operatorname{div}(v)) \nabla I + \lambda(H I v + (Dv)^T \nabla I)
\tag{7.27}
$$

$$
+ \quad \sum_{i=0}^{N-1} \Big[ (\nabla \gamma_i)^T v + \gamma_i \operatorname{div}(v) - G_\sigma \star (m^T \nu_i) \Big] \nabla h_i + \gamma_i (H h_i v + (Dv)^T \nabla h_i)
\tag{7.28}
$$

$$
= \quad (\lambda \nabla I + \sum_{i=0}^{N-1} \gamma_i \nabla h_i) \operatorname{div}(v) + (Dv)^T [\lambda \nabla I + \sum_{i=0}^{N-1} \gamma_i \nabla h_i] + (\nabla \lambda^T v) \nabla I
\tag{7.29}
$$

$$
+ \quad \lambda H I v + \sum_{i=0}^{N-1} \Big[ (\nabla \gamma_i)^T v \Big] \nabla h_i + \gamma_i H h_i v - G_\sigma \star (m^T \nu_i) \nabla h_i \,.
\tag{7.30}
$$

Noticing that

$$
D(\lambda \nabla I) v = \lambda H I v + \nabla \lambda^T v \nabla I
\tag{7.31}
$$

we can write

$$
(\nabla \lambda^T v) \nabla I + \lambda H I v + \sum_{i=0}^{N-1} ((\nabla \gamma_i)^T v) \nabla h_i + \gamma_i H h_i v
\tag{7.32}
$$

$$
= D(\lambda \nabla I) v + \sum_{i=0}^{N-1} D(\gamma_i \nabla h_i) v = (Dm) v \,.
\tag{7.33}
$$

Finally, we get

$$
-m_t = m \operatorname{div}(v) + (Dv)^T m + (Dm) v - \sum_{i=0}^{N-1} G_\sigma \star (m^T \nu_i) \nabla h_i \,,
\tag{7.34}
$$

which gives the result.

## 7.3 Energy Conservation

To formulate a shooting-based solution we would like to avoid integrating $\|v\|_L^2$ over time. We here show that this quantity is conserved. Hence, $\int_0^1 \|v\|_L^2 \, \mathrm{d}t = \|v(0)\|_L^2$, which allows us to write our shooting equations only with respect to initial conditions subject to the momentum-based evolution equations of RDMM.

Recall that the energy is preserved by the EPDiff equation since it can alternatively be written as

$$\partial_t m + \mathrm{ad}_v^* m = 0 \,, \tag{7.35}$$

where $\mathrm{ad}^*$ is the adjoint of $\mathrm{ad}_v \, w := \mathrm{d}v(w) - \mathrm{d}w(v)$. It implies that

$$\frac{\mathrm{d}}{\mathrm{d}t}\langle m, K \star m \rangle = -2\langle \mathrm{ad}_v^* m, K \star m \rangle = \langle m, \mathrm{ad}_v \, v \rangle = 0 \,,$$

since $\mathrm{ad}_v \, v = 0$. In fact, there is more than conservation of the energy, since the momentum is actually advected along the flow. Now, formula (3.9) can be shortened as $\partial_t m + \mathrm{ad}_v^* m = \sum_i G_\sigma \star (m^T \nu_i)\nabla h_i$ and it implies that, denoting the kernel $K(w_i)$ to shorten the notations,

$$\frac{\mathrm{d}}{\mathrm{d}t}\langle m, K(w_i) \star m \rangle = -2\langle \mathrm{ad}_v^* m, K(w_i) \star m \rangle + 2\langle \sum_i G_\sigma \star (m^T \nu_i)\nabla h_i, v \rangle + 2\langle m, \sum_i \partial_t w_i \nu_i \rangle$$

$$= 2\langle \sum_i G_\sigma \star (m\nu_i)\nabla h_i, v \rangle + 2\langle m, \sum_i (\partial_t w_i)\nu_i \rangle = 0$$

since the first term vanishes as for the standard EPDiff equation and the two other terms cancel each other since $\partial_t w_i = -G_\sigma \star \nabla h_i \cdot v$ and $v = \sum_i G_{\sigma_i} \star (w_i m)$. Here we assumed the kernel to be symmetric in writing this equation but the result holds in general, the equations being simply modified with the transpose kernel.

## 7.4 Mathematical properties

In this section, we prove that given $(\nu_i)_{i=0,\ldots,N-1}$, there exists a solution $\varphi(t)$ solving Equations (7.7) and (7.8) at least until a time $T > 0$ which could be less than 1. The notations $\|\cdot\|_{k,\infty}$ or $\|\cdot\|_{C^k}$ denote the sup norm of $C^k$ maps.

**Theorem 3.** *Let $V_{N-1} \subset \ldots \subset V_0$ and suppose for every $\nu_k \in V_k$, $\|\nu_k\|_{V_k} \le const\|\nu_k\|_{V_1} \le const\|\nu_k\|_{2,\infty}$. Given initial weights $(h_i(t=0))_{i=0}^{N-1} \in L^2$ and time dependent vector fields $\nu_i(t) \in V_i$, there exists a unique solution $\varphi(t)$ to Equations (3) until time 1.*

*Proof.* The first step consists in proving that there exists a solution locally in time. To this end, the proof follows a fixed point argument on the space $C^0([0,1], \mathrm{Diff}_{C^1}(\Omega))$, i.e. the space of continuous curves in $\mathrm{Diff}_{C^1}(\Omega)$, for the map

$$T(\varphi) := \mathrm{Fl}(v) \tag{7.36}$$

where $v$ is defined as $v[\varphi] := \sum_{i=0}^{N-1} G_\sigma \star h_i(\varphi^{-1}(t,y))\nu_i(t,\varphi(t,x))$. The existence of the flow associated with $v[\varphi]$ is ensured by standard arguments provided that the Lipschitz constant of $v[\varphi]$ is bounded. It is the case since $G_\sigma(x,y) \star h_i(\varphi^{-1}(t,y))$ has a Lipschitz constant bounded by $\sup_{x \in \Omega} |\partial_1 G_\sigma(x,y)|$ since $|h_i(\varphi^{-1}(t,y))| \le 1$. This gives $\|v[\varphi]\|_{1,\infty} \le \sum_{i=0}^{N-1} M\|\nu_i\|_{1,\infty}$ and the constant $M$ does not depend on $\varphi$. A similar inequality holds for the sup norm on the derivatives up to order $k$ provided each space $V_i$ continuously embeds in $C^k$.

One has the inequality

$$\|T(\varphi)(t)\|_{2,\infty} \le e^{\int_0^t \|v[\varphi]\|_{2,\infty} \, \mathrm{d}s} \tag{7.37}$$

and therefore, $\|T(\varphi)(t)\|_{2,\infty}$ is bounded a priori by a positive constant which does not depend on $\varphi$. Using Gronwall's lemma (6), one has also

$$\|T(\varphi)(t) - T(\psi)(t)\|_{1,\infty} \le \sqrt{t}\|v[\varphi] - w[\psi]\|_{L^2([0,t],C^1)} e^{\int_0^t (1+\|\varphi\|_{1,\infty})\|v\|_{C^2} \, \mathrm{d}s} \,. \tag{7.38}$$

Moreover, by a change of variable $y = \varphi(t,x)$ we have

$$G_\sigma \star h_i(\varphi^{-1}(t,y)) = G_\sigma(x,\varphi(t,y)) \star (\mathrm{Jac}(\varphi(t,y))h_i(y)) \tag{7.39}$$

and therefore

$$\|G_\sigma \star h_i(\varphi^{-1}(t,y))\nu_i - G_\sigma \star h_i(\psi^{-1}(t,y))\nu_i\|_{C^1} \le M'\|\varphi - \psi\|_{C^1}\|\nu_i\|_{C^1} . \qquad (7.40)$$

Thus, we deduce the inequality

$$\|v[\varphi] - w[\psi]\|_{L^2([0,t],C^1)} \le M \sup_{s \in [0,t]} \|\psi(s) - \varphi(s)\|_{C^1} , \qquad (7.41)$$

therefore, the map $T$ is a contraction for a time $T$ small enough. Using Equation (7.37), it is easily seen that this existence can be applied on $[T, 2T]$ and iterating this argument shows existence until time $t = 1$. $\qquad \square$

**Theorem 4.** *The variational problem* (7.7) *under the constraints of Equations* (7.8) *has a solution.*

*Proof.* The direct method of calculus of variations can be applied here, see Sect. 7.2. The sum of squared norms are lower semicontinuous; The penalty term as well as the constraints are weakly closed for the weak convergence on $(\nu_i)$. $\qquad \square$

**Lemma 5.** *Let $u, v \in L^2([0,1], C^2)$ and let $\varphi, \psi$ be their associated flows. The following estimates hold,*

$$\|\varphi(t)\|_{C^2} \le e^{\int_0^t \|v(s)\|_{C^2} \, ds} , \qquad (7.42)$$

*and*

$$\|\varphi(t) - \psi(t)\|_{C^1} \le \sqrt{t}M\|u - v\|_{L^2([0,t],C^1)}e^{\int_0^t (1+\|\varphi\|_{1,\infty})\|v\|_{C^2} \, ds} . \qquad (7.43)$$

*where $M$ is a constant that bounds $\|\varphi\|_{1,\infty}$.*

*Proof.* Use Gronwall's lemma (6) recalled below on the following inequality coming from the flow equation

$$\|\varphi(t) - \psi(t)\|_{1,\infty} \le \int_0^t \| \, du \circ \varphi(t) \cdot d\varphi(t) - dw \circ \psi(t) \cdot d\psi(t)\|_{0,\infty} \, ds \qquad (7.44)$$

$$\le \int_0^t \|u - v\|_{1,\infty}\|\varphi\|_{1,\infty} + \|v\|_{2,\infty}\|\varphi\|_{1,\infty}\|\varphi - \psi\|_{0,\infty} + \|dv\|_{1,\infty}\|\varphi - \psi\|_{1,\infty} \, ds \qquad (7.45)$$

$$\le \int_0^t \|u - v\|_{1,\infty}\|\varphi\|_{1,\infty} + (1 + \|\varphi\|_{1,\infty})\|v\|_{2,\infty}\|\varphi - \psi\|_{1,\infty} \, ds . \qquad (7.46)$$

$$\square$$

Recall that Gronwall's lemma is

**Lemma 6.** *Let $r$ be a nonnegative function on $\mathbb{R}$ such that*

$$r(t) \le c(t) + \left| \int_0^t \alpha(s)r(s) \, ds \right| \qquad (7.47)$$

*for given positive functions $\alpha$ and $c$. Then,*

$$r(t) \le c(t) + \left| \int_0^t \alpha(s)c(s)e^{| \int_0^t \alpha(s) \, ds |} \, ds \right| , \qquad (7.48)$$

*and if $c$ is a constant, a further simplified formula is*

$$r(t) \le ce^{| \int_0^t \alpha(s) \, ds |} . \qquad (7.49)$$

Figure 8: Flow chart of LDDMM (left) and our RDMM model (right). LDDMM solves EPDiff and advects the transformation map, whereas in RDMM a modified EPDiff equation is solved combined with an advection of the transformation map and the pre-weights for the regularizer. Note that transformation map $\varphi^{-1}$ and the pre-weights $\{h_i\}$ are both advected according to the velocity field. Hence, instead of computing the advection equation, we update $\{h_i(t)\}$ by interpolating the initial pre-weights $\{h_i(0)\}$ via the current transformation map $\varphi^{-1}(t)$, which is more computationally efficient and avoids numerical dissipation.

## 7.5 Initial Pre-weight Regularization

The initial regularization term $Reg(\{h_i(0)\})$ determines the behavior of the initial regularizer. In our experiments,

$$Reg(\{h_i(0)\}, T) = \lambda_{\mathrm{OMT}}(T)OMT(\{h_i(0)\}) + \lambda_{\mathrm{Range}}(T)Range(\{h_i(0)\}), \qquad (7.50)$$

where $\lambda_{\mathrm{OMT}}$ and $\lambda_{\mathrm{Range}}$ are scale factors; T refers to the iteration/epoch. Specifically,

$$OMT = \left|\log\frac{\sigma_{N-1}}{\sigma_0}\right|^{-s} \sum_{i=0}^{N-1} w_i \left|\log\frac{\sigma_{N-1}}{\sigma_i}\right|^s \qquad (7.51)$$

where $s$ is the chosen power and

$$Range = \|G_\sigma \star (h(0)) - w_0\|_2^2 \qquad (7.52)$$

where $w_0$ is the pre-defined initial weight. The range loss penalizes differences between the initial weight, $w(0)$, from the pre-defined one, $w_0$.

At the beginning of the optimization/training, it is difficult to jointly optimize over the momentum and the pre-weights. Hence, we constrain the pre-weights by introducing the Range loss that penalizes the difference between the optimized and pre-defined pre-weights. Besides, as we prefer well-regularized (i.e., smooth) transformation, we use the OMT loss to penalize weight assignments to Gaussians with small standard deviations. To solve the original model, the influence of the range penalty needs to diminish while the influence of the OMT term need to increase during training. In practice, we introduce epoch-dependent decay factors:

$$\lambda_T = \frac{K}{K + e^{T/K}} \quad , \lambda_{\mathrm{Range}} := C_{\mathrm{Range}}\lambda_T, \quad \lambda_{\mathrm{OMT}} = C_{\mathrm{OMT}}(1 - \lambda_T), \qquad (7.53)$$

where $C_{\mathrm{Range}}$ and $C_{\mathrm{OMT}}$ are pre-defined constants, $K$ controls the decay rate, and $T$ indicates the iteration/epoch.

## 7.6 LDDMM/RDMM Unit

Fig. 8 shows the flow charts for LDDMM and RDMM. Additionally, for RDMM with a pre-defined regularizer, we define $\{h_i(0)\}$ in the source image space, as foreground and background are easier to specify there. For RDMM with an optimized/learnt regularizer we define $\{h_i(0)\}$ in the pre-aligned image space, since the goal is to find the optimal initial conditions that determine the geodesic path based on the RDMM shooting equations; specifically, we take $\varphi^{-1}(0) = id$ as the input, and the final output composes the initial map and the transformation map, $\varphi^{-1}(1)$.

## 7.7  Analysis of the OMT term

This section illustrates the behavior of the OMT term to obtain regular solutions. To understand the behavior of the OMT term, we do some simple analysis. We assume

$$0 < \sigma_0 < \sigma_1 < \cdots < \sigma_{N-1} \tag{7.54}$$

where $\sigma_i$ are the standard deviations of the Gaussians. In general, we desire

$$c_0 > c_1 > \cdots > c_{N-1}, \tag{7.55}$$

where $c_i$ are the associated costs of assigning a weight to $i$-th Gaussian. That is, we assume that it gets progressively cheaper to assign to Gaussians with larger standard deviation. However, we do not assume that this is the case in the following derivations. Our OMT penalty term is then of the form

$$f(w) = \sum_{i=0}^{N-1} c_i w_i = c^\top w \tag{7.56}$$

with constraints

$$\sum_{i=0}^{N-1} w_i - 1 = 1^\top w - 1 = 0 \quad \text{and} \quad w_i \geq 0 \ . \tag{7.57}$$

To study this term, assume that we have a given target standard deviation $\hat{\sigma}$ that we wish to explain via a multi-Gaussian. This results in the constraint

$$\sum_{i=0}^{N-1} \sigma_i^2 w_i - \hat{\sigma}^2 = v^\top w - \hat{\sigma}^2 = 0 \ . \tag{7.58}$$

This optimization problem is linear in the multi-Gaussian weights, $w$, and consequentially constitutes the following linear program

$$\min_{w} f(w) \quad \text{s.t.} \quad \begin{cases} 1^\top w - 1 & = 0, \\ v^\top w - \hat{\sigma}^2 & = 0, \\ w_i & \geq 0 \ . \end{cases} \tag{7.59}$$

The Lagrangian of this problem is

$$L(w, \lambda, \gamma_1, \gamma_2) = f(w) - \lambda^\top w - \gamma_1 (1^\top w - 1) - \gamma_2 (v^\top w - \hat{\sigma}^2) \ , \tag{7.60}$$

which results following KKT conditions [24]

$$c - \lambda - \gamma_1 1 - \gamma_2 v = 0, \tag{7.61}$$
$$1^\top w - 1 = 0, \tag{7.62}$$
$$v^\top w - \hat{\sigma}^2 = 0, \tag{7.63}$$
$$w \geq 0, \tag{7.64}$$
$$\lambda \geq 0, \tag{7.65}$$
$$\lambda_i w_i = 0, \ \forall \, i \ . \tag{7.66}$$

### 7.7.1  Solution on a simplex edge

Assume a solution candidate for the KKT conditions (Eqs. (7.61)-(7.66)) that only has two zero weights

$$w_k > 0, \ w_l > 0, \ w_i = 0 \ \forall i \notin \{k, l\}, \ \sigma_k < \sigma_l \ . \tag{7.67}$$

Then, we know

$$w_k + w_l = 1, \quad \sigma_k^2 w_k + \sigma_l^2 w_l = \hat{\sigma}^2, \quad \lambda_k = \lambda_l = 0 \ . \tag{7.68}$$

Using Eq. (7.68), we can directly solve for $w_k$ and $w_l$ and obtain

$$w_k = \frac{\sigma_l^2 - \hat{\sigma}^2}{\sigma_l^2 - \sigma_k^2}, \quad w_l = \frac{\hat{\sigma}^2 - \sigma_k^2}{\sigma_l^2 - \sigma_k^2} \ . \tag{7.69}$$

Note that these weights are independent of the costs $c_i$[4].

Both weights are required to be non-negative, *i.e.* $w_k \geq 0, w_l \geq 0$. From this condition, we obtain

$$w_k \quad = \quad \frac{\sigma_l^2 - \hat{\sigma}^2}{\sigma_l^2 - \sigma_k^2} \geq 0 \tag{7.70}$$

$$\Leftrightarrow \quad \sigma_l^2 - \hat{\sigma}^2 \geq 0 \tag{7.71}$$

$$\Leftrightarrow \quad \hat{\sigma}^2 \leq \sigma_l^2 \tag{7.72}$$

and

$$w_l \quad = \quad \frac{\hat{\sigma}^2 - \sigma_k^2}{\sigma_l^2 - \sigma_k^2} \geq 0 \tag{7.73}$$

$$\Leftrightarrow \quad \hat{\sigma}^2 - \sigma_k^2 \geq 0 \tag{7.74}$$

$$\Leftrightarrow \quad \sigma_k^2 \leq \hat{\sigma}^2 \tag{7.75}$$

and finally that the desired variance needs to be between the variances of $k$ and $l$, i.e., $\sigma_k^2 \leq \hat{\sigma}^2 \leq \sigma_l^2$. Since $\lambda_k = \lambda_l = 0$, we further get from Eq. (7.61) that

$$c_k - \gamma_1 - \gamma_2 \sigma_k^2 \quad = \quad 0 \;, \tag{7.76}$$

$$c_l - \gamma_1 - \gamma_2 \sigma_l^2 \quad = \quad 0 \;, \tag{7.77}$$

which we can solve for $\gamma_1$ and $\gamma_2$ to obtain

$$\gamma_1 = \frac{c_k \sigma_l^2 - c_l \sigma_k^2}{\sigma_l^2 - \sigma_k^2}, \quad \gamma_2 = \frac{c_l - c_k}{\sigma_l^2 - \sigma_k^2} \;. \tag{7.78}$$

For arbitrary $i \notin \{l, k\}$ the Lagrangian multipliers are then

$$\lambda_i \quad = \quad -\gamma_1 - \gamma_2 \sigma_i^2 + c_i, \tag{7.79}$$

$$= \quad \frac{\sigma_l^2(c_i - c_k) + \sigma_k^2(c_l - c_i) + \sigma_i^2(c_k - c_l)}{\sigma_l^2 - \sigma_k^2}, \tag{7.80}$$

$$= \quad \frac{c_i(\sigma_l^2 - \sigma_k^2) + c_k(\sigma_i^2 - \sigma_l^2) + c_l(\sigma_k^2 - \sigma_i^2)}{\sigma_l^2 - \sigma_k^2} \;. \tag{7.81}$$

Since $\lambda_i \geq 0$ we get

$$c_i \geq c_k \frac{\sigma_l^2 - \sigma_i^2}{\sigma_l^2 - \sigma_k^2} + c_l \frac{\sigma_i^2 - \sigma_k^2}{\sigma_l^2 - \sigma_k^2} = g(\sigma_i^2). \tag{7.82}$$

As $g(\sigma_k^2) = c_k$ and $g(\sigma_l^2) = c_l$, this is simply a line that passes through the points $(\sigma_k^2, c_k)$ and $(\sigma_l^2, c_l)$ and this condition states that for a solution candidate edge $(k, l)$ the costs for all $i \notin \{k, l\}$ are on or above this line. If the costs are defined via a function $c = h(\sigma^2)$, then, if $h$ is a convex function (and remembering the condition $\sigma_k^2 \leq \hat{\sigma}^2 \leq \sigma_l^2$), the optimal solution of this linear program will be on the edge $(k_*, l_*)$ most tightly bracketing $\hat{\sigma}^2$, i.e.,

$$\sigma_{k_*}^2 \leq \hat{\sigma}^2 \leq \sigma_{l_*}^2, \quad \text{s.t.} \quad k_* = \max_i \{i : \sigma_i^2 \leq \hat{\sigma}^2\}, \; l_* = \min_i \{i : \hat{\sigma}^2 \leq \sigma_i^2\} \;. \tag{7.83}$$

### 7.7.2 Solution on a simplex vertex

Assume that $\hat{\sigma}^2 = \sigma_j^2$, i.e., the desired variance coincides with the variances of one of the Gaussians. Then $w_j = 1$ and $w_i = 0$, $\forall i \neq j$. Furthermore, we have $\lambda_j = 0$ from which follows

$$\gamma_1 = c_j - \gamma_2 \sigma_j^2 \;. \tag{7.84}$$

For the remaining $N - 1$ variables $i \neq j$, it needs to hold that

$$c_i - \lambda_i - \gamma_1 - \gamma_2 \sigma_i^2 = 0 \;. \tag{7.85}$$

Substituting Eq. (7.84), we can solve for $\lambda_i$ and obtain

$$\lambda_i = c_i - c_j + \gamma_2(\sigma_j^2 - \sigma_i^2) \ . \tag{7.86}$$

As all the Lagrangian multipliers, $\lambda$, are (by the KKT conditions) required to be non-negative, we obtain the condition

$$c_i - c_j + \gamma_2(\sigma_j^2 - \sigma_i^2) \geq 0 \ . \tag{7.87}$$

We can distinguish two conditions

$$\gamma_2 \quad \geq \quad \frac{c_j - c_i}{\sigma_j^2 - \sigma_i^2}, \text{ for } i < j \ , \tag{7.88}$$

$$\gamma_2 \quad \leq \quad \frac{c_k - c_j}{\sigma_k^2 - \sigma_j^2}, \text{ for } j < k \ . \tag{7.89}$$

Hence, it needs to hold that

$$\frac{c_j - c_i}{\sigma_j^2 - \sigma_i^2} \leq \frac{c_k - c_j}{\sigma_k^2 - \sigma_j^2}, \ \forall i < j < k \ . \tag{7.90}$$

Since $\sigma_i^2 < \sigma_j^2 < \sigma_k^2$, we can multiply by $(\sigma_j^2 - \sigma_i^2)(\sigma_k^2 - \sigma_j^2)$ and obtain

$$(c_j - c_i)(\sigma_k^2 - \sigma_j^2) \leq (c_k - c_j)(\sigma_j^2 - \sigma_i^2) \ . \tag{7.91}$$

Solving for $c_j$, we get

$$c_j \leq c_i \frac{\sigma_k^2 - \sigma_j^2}{\sigma_k^2 - \sigma_i^2} + c_k \frac{\sigma_j^2 - \sigma_i^2}{\sigma_k^2 - \sigma_i^2} \ . \tag{7.92}$$

Now, for a vertex $j$ to be a solution, this condition needs to be true for all $i < j < k$. Graphically, we can take any two points $(\sigma_i^2, c_i)$ and $(\sigma_k^2, c_k)$ and draw a line between them. If $(\sigma_{k,j}^2)$ is below these lines for all pairs $(i, k)$ such that $i < k < j$, then there is a vertex solution, otherwise the solution is on an edge (for non-degenerate $c_i$). This condition is always fulfilled for convex functions $c = h(\sigma^2)$.

### 7.7.3 Solution on a simplex face

We can also ask if it is possible that a solution is on a face of the simplex. Consider the case of three non-zero weights $w_k > 0$, $w_l > 0$, $w_m > 0$. In this case, we have

$$w_k + w_l + w_m \quad = \quad 1 \ , \tag{7.93}$$

$$\lambda_k = \lambda_l = \lambda_m \quad = \quad 0 \ , \tag{7.94}$$

$$\sigma_k^2 w_k + \sigma_l^2 w_l + \sigma_m^2 w_m \quad = \quad \hat{\sigma}^2 \ . \tag{7.95}$$

Furthermore, to fulfill the KKT conditions, the following equation system needs to hold (and similarly for more than three non-zero weights):

$$\begin{pmatrix} c_k \\ c_l \\ c_m \end{pmatrix} = \begin{pmatrix} 1 & \sigma_k^2 \\ 1 & \sigma_l^2 \\ 1 & \sigma_m^2 \end{pmatrix} \begin{pmatrix} \gamma_1 \\ \gamma_2 \end{pmatrix} . \tag{7.96}$$

For more than two non-zero weights this is an overdetermined system. Hence, a solution can only be on a face if this system has a solution. In that case, this means that

$$c_m = c_l \frac{\sigma_k^2 - \sigma_m^2}{\sigma_k^2 - \sigma_l^2} + c_k \frac{\sigma_m^2 - \sigma_l^2}{\sigma_k^2 - \sigma_l^2} \ . \tag{7.97}$$

In other words, this is only possible if $c = h(\sigma^2)$ is at least piecewise linear. Specifically, if $c = h(\sigma^2)$ is a *strictly* convex function, a solution can only be found on simplex edges or vertices based on the conditions in the two previous sections.

### 7.7.4 Summary

In summary, to assure that a solution exists either 1) based on the two Gaussians closest to the desired variance $\hat{\sigma}^2$ (i.e., a simplex edge) or 2) selecting exactly one of the Gaussians (a simplex vertex), it is desirable to pick a penalty function based on costs from a strictly convex function $c = h(\sigma^2)$. Based on our choice for the OMT penalty:

$$c = h(x) = \frac{1}{2^r} \left( \log \frac{\sigma_{N-1}^2}{x} \right)^r , \qquad (7.98)$$

we get

$$\frac{d^2 h(x)}{dx^2} = \frac{\frac{r}{2^r} \log^{r-2} \left( \frac{\sigma_{N-1}^2}{x} \right) \left( \log \frac{\sigma_{N-1}^2}{x} + r - 1 \right)}{x^2}. \qquad (7.99)$$

For $0 < x < \sigma_{N-1}^2$ and $r \geq 1$, the second derivative is positive and hence $h(x)$ is strictly convex.

### 7.8 Experimental settings

For all experiments, we normalize the intensities of each image such that the $0.1$th percentile and the $99.9$th percentile are mapped to $0, 1$ and clamp values that are smaller to $0$ and larger to $1$ to avoid outliers. We also assume that spatial coordinates of images are in $[0, 1]^d$, where $d$ is the spatial dimension. This makes the interpretation of the standard deviations of the regularizers straightforward.

**Non-parametric family** For numerical optimization solutions, we use three image scales $\{0.25, 0.5$ and $1.0\}$. We use L-BGFS as the optimizer. For the deep learning models, we train the multi-step affine network first and then train the non-parametric network with the affine network fixed. For all methods, we use a multi-kernel Gaussian regularizer with standard deviations $\sigma_i = \{0.05, 0.1, 0.15, 0.2, 0.25\}$. For both vSVF and LDDMM, we use fixed corresponding weights $w_0^2 = \{0.067, 0.133, 0.2, 0.267, 0.333\}$, which is also set as the initial value for the range loss in RDMM.

**Baseline methods**

For the numerical optimization approaches, we compare with three public registration tools: NiftyReg [25, 20, 30, 21], SyN [2, 1] and Demons [38, 37]. Besides, we also compare with two recent deep-learning approaches: VoxelMorph [10] and AVSM (vSVF-net)[33]. For a fair comparison, we take the same experimental settings as in [33].

**Registration with a pre-defined regularizer**

For the synthetic registration experiments, we use a multi-kernel Gaussian regularizer with standard deviations $\sigma_i = \{0.03, 0.06, 0.09, 0.3\}$ with initial pre-weights $h_0^2 = \{0.2, 0.5, 0.3, 0.0\}$ (fixed during the optimization) for the foreground (the dark blue region) and $h_0^2 = \{0, 0, 0, 1\}$ for the background (the cyan region). The standard deviation for $G_\sigma$, to smooth the pre-weights, is set to $0.02$. For each image scale, we compute $60$ registration iterations.
For the lung registration we use $\sigma_i = \{0.04, 0.06, 0.08, 0.2\}$ for the multi-Gaussian regularizer with initial pre-weights $h_0^2 = \{0.1, 0.4, 0.5, 0\}$ for the foreground (i.e.the lung) and $h_0^2 = \{0, 0, 0, 1\}$ everywhere outside the lung. We set $\sigma$ in $G_\sigma$ to $0.05$. For each image scale, we compute $60$ registration iterations.

**Registration with an optimized regularizer**

For the synthetic registration experiment, we use $\sigma_i = \{0.02, 0.04, 0.06, 0.08\}$ for the multi-Gaussian regularizer and the initial pre-weights $h_0^2 = \{0.1, 0.3, 0.3, 0.3\}$. These initial value are set at the beginning of the optimization. They are the same for vSVF, LDDMM and RDMM. $C_{\text{Range}}$ and $C_{\text{OMT}}$ are set to $10$ and $0.05$ respectively; K is set to $10$; $\sigma$ in $G_\sigma$ is set to $0.05$. For image scale $\{0.25, 0.5$ and $1.0\}$, we compute $\{100, 100$ and $400\}$ iterations, respectively.
For the knee MRI registration of the OAI dataset, we use $\sigma_i = \{0.05, 0.1, 0.15, 0.2, 0.25\}$ for the multi-Gaussian regularizer with the initial pre-weights $h_0^2 = \{0.067, 0.133, 0.2, 0.267, 0.333\}$ for all non-parametric registration models. $C_{\text{Range}}$ and $C_{\text{OMT}}$ are set to $1$ and $0.25$ respectively; K is set to $6$; $\sigma$ in $G_\sigma$ is set to $0.06$. For each image scale, we compute $60$ registration iterations.

**Registration with a learnt regularizer**

For the deep learning approaches, the settings are the same as for numerical optimization, as described above. Besides, we include an additional inverse consistency loss, with the scale factor set to 1e-4, for vSVF, LDDMM and RDMM to regularize the deformation.

## Footnotes

[2]We multiply the objective function by $\frac{1}{2}$ for convenience. This does not change the solution.

[3]In this section, to simplify the notation, we denote the partial derivative $\partial_t$ by only the subscript $_t$.

[4]We will show in the remainder of this section that for costs defined via a convex function, $g$, $c = g(\sigma^2)$, this is indeed a solution of the KKT equations. This is consistent with known optimal mass transport theory, where for convex costs the optimal mass transport in 1D is a monotone rearrangement [12].