[Reviews · NeurIPS 2019]

Reviewer 1



The paper is well written, although *very* dense both in terms of mathematical expectation and development, as well as in terms of space. It is *not* an easy read (I suppose unless you are super fluent in LDDMM background). I think the authors could improve this to help this paper reach a broader audience, or perhaps they are not interested in this, I'm not sure. As it stands, it is a *bit* hard to evaluate due to the super condensed and dense nature of it. I believe this is a technical clean contribution with a clear advancement. However, I have a few concerns, but am happy to read and evaluate a rebuttal. The more serious concerns are with the experiments, below. General conceptual concerns: - In general, the usefulness of spatially varying regularization is confusing to me. I know there is literature towards this end, but in some sense regularization can be seen as a prior, and a prior is devoid of effects from data. In a bayesian perspective, wouldn't it be up to the data to affect the deformation such that it "overcomes" the prior when the signal is strong enough? Other than technical curiosity, could the authors argue the need for this -- or make it clear that it's a technical exercise? - Along the same lines, while following all of the LDDMM development, I've never seen a practical example where the time-dependent velocities/momenta provide any advantage over stationary ones, but perhaps the authors could provide an example (since their model is time-dependent). It's also fine if this is a technical exercise, but it would be great if the authors are up-front about this. - I don't fully follow how the velocity field is bounded at the boundaries of segmentations (line 113), and in fact the experiments seem to have more folding voxels there (as the authors state). Can the authors explain if this is, indeed true? In fact, it's unclear to me why invertibility needs to be maintained -- wouldn't this method be applicable in situations where there are strong changes along edges where smoothness and invertibility is not necessary? - I am a bit confused why the symmetric loss (169) is necessary -- wouldn't the fact that the deformations are invertible guarantee this property? Is the loss aimed at regularizing numerical issues, in a sense? When using the DL frameworks, the numerical approximation should be good enough to avoid this issue. - Finally, I am a bit unclear if this belongs in NeurIPS. I believe this could be a very appropriate (and strong) contribution in a place like MICCAI or even directly to IEEE-TMI or similar. I am not sure, however, of how the development applies to NeurIPS -- there are minimal contributions to learning, neuroscience or optimization, and the main contribution is the extension/improvement to LDDMM theory, which would probably fit more in the differential geometry or, as the authors state, fluid mechanics. However, perhaps I am misinterpretting the growing encapsulation of NeurIPS -- I welcome an author rebuttal and Meta-Reviewer override on this aspect. Experimental concerns: - For the DL approaches: In the reproducibility checklist, there are questions about ensuring that the data is properly split into train/validate/test sets, and that there is sufficient description about motivation of hyper-parameter choices. These questions are really important in ML -- without a separate validation set (separate from the test set, which should be held out), for example, it is very likely that the hyper-parameters are over-fit. While the authors responded 'yes' to these questions, these aspects re missing from the work -- as far as I can tell, there is no validation/test set discussion. I understand that this is registration and some of these methods are unsupervised, but it's been shown that there is a difference between train and test set of these networks depending on the amount of training data. What datasets are the results on ? Why does this differ from the checklist? - To me, the results are very similar to the baselines -- e.g. LDDMM and the DL methods. While the authors bold their results, as far as I can see the results are essentially the same -- if the authors disagree, can they provide a statistical significance test? If not, the practical advantage/utility of the method seems quite limited -- RDMM shows more folds (minimal, but more), and the maps they produce are, I would argue, only minimally informative -- in fact, I suspect that if the deformation fields themselves are visualized in image-based ways, the same or similar features can also be seen. Perhaps the authors can explain how I am wrong in this regard -- but it does seem that overall, the results are comparable (in accuracy, regularity (folds), runtime, and insight (if visualizing the deformation)). Unless there is a strong counter-argument to this, I would have liked to see a more clear write-up essentially saying: "Look, our contribution is mainly theoretical improvement, and these experiments are mostly to validate that we don't break anything compared to current approaches. We don't mean that this by itself will be useful, but rather that it will provice theoretical grounding for the future" -- I think this would make it more clear what the setting of the paper is. As it stands, it seems like at times the authors try to declare that RDMM is better in some ways, but I personally don't observe this. Again, I'm happy to read a rebuttal to this. - Less important -- I'm curious why the authors didn't apply this work to brain, which is probably the most often use case that I've seen used with LDDMM, and this would fit well with NeurIPS since there is an entire neuroscience sub-community. Minor: - Please be careful about the order of introducing mathematical definitions. For example, you define <.,.> as inner product after having used it several times already. - Please specify where in the suppl. Material to refer to for each part where you say "(see suppl. Material)". It's very hard to track otherwise. - Please cite papers correctly -- [10] is MICCAI, [11] is DLMIA, [17] is ICLR, [23] and [33] are CVPR, etc. - Considering the NeurIPS now accepts code at submission, stating "our code will be open-sourced" is peculiar (and if anything, raises skepticism for me). All papers can make this empty promise -- if you'd like that to be a bonus to help your paper, include it in the submission. * update after author response: Thank you for the thorough response. Overall, I don't feel like I should change my overall score -- I *do* certainly still feel that the paper should be accepted, and will argue for this in the discussion, but currently feel that a 7 is a fair assessment. I believe the authors mainly addressed the train/validate/test split issue, but several concerns still stand -- the answers for several of them were a bit too short for me to understand how they are really addressed (e.g. the symmetry question)..

Reviewer 2



The material of this submission is substantial. Unfortunately, the NeurIPS template is short and the author have postponed many details in the appendix. Splitting the paper in two part has a drawback: it makes the reading not as smooth as it could be in a single file as the Appendix now contains large technical blocks (with few text) making it tedious to read. The paper contains few typos and the proof seems to me correct even if I don't check everything line by line. Here are the minor typos I found: Paper - lines 4-8 (Abstract): The sentence is too long. Please consider to split it in two. - page 2 (caption Fig 1): t=0 -> $t=0$ and t=1 -> $t=1$ - line 92: missing a space after w.r.t. - lines 107 and 372: Is it possible to find a better notation than $K = \sum_i w_i K w_i$? - equation 3.5: Please introduce notation \frac{\delta}{\delta I(1)}. supplementary material - Equation 7.1 : there is an extra | (vert) in LHS. - Equation 7.5: the dependence on $t$ is implicit in RHS. Please make it explicit. - line 421: euqation -> equation

Reviewer 3



The paper is however hard on notation, possibly restricting a clear reading. One example is a lack of formal definition of used term, such as Reg(), cramping many notation in single lines (eq 2.1-2.4), heavy wording ("regularize the regularizer"). For this reason, perhaps, I missed an important step in the paper: How are the initial momenta exactly learned? Is this from ground truth momenta, if so, are they commuted via a conventional gradient-descent optimization? If so, (a) how does the learning method compares with the learned method, and (b) how does using the conventional optimization method compares with non-spatially-varying methods (LDDMMs, SVFs). In the experiment, the proposed learning method is directly compared with conventional non-spatially-varying method, which prevents the reader to appreciate the contribution from (a) the learning approach, and (b) the spatially-varying regularization. In other word, is deep learning really adding value to the registration regularization (the contribution)? Spatially-varying method has been studied in the medical imaging community, notably with the work on "Probabilistic non-linear registration with spatially adaptive regularization", MICCAI'13, MedIA'15. Other follow up work should be mentioned and studied if the spatially-varying aspect is promoted in the submission.

[Author Response · NeurIPS 2019]

We thank the reviewers for their efforts in reviewing and their feedback, considering the paper "well written", "[a
technically] clean contribution with a clear advancement" as well as "clear, rigorous, and self-contained". Furthermore,
R2 commented that our "methods mixing geometric deformation models and DL may interest the NeurIPS community."

**The main reviewer concerns revolved around (see also additional responses below):**

**1.** *Modeling assumptions (R1/R4).* Our model is based on spatio-temporal velocity fields (rather than stationary velocity
fields (SVF)), because this allows us to *preserve theoretical properties.* Importantly, no registration approaches exist
which model and estimate metrics which jointly deform with space. Our model, containing LDDMM as a special case,
is therefore *a first theoretical contribution in this direction, demonstrated on a practical large-scale registration task.*

**2.** *The overall formulation, contribution and performance of our approach (R1/R4).* Our approach is related to [23] in
the sense that we estimate a metric, but we do so in an LDDMM (not SVF) setting. Further, as in our approach (but not
for [23]) the metric moves with the deforming space *we derived an entirely new registration model (RDMM).* While our
model can be estimated via standard optimization, we use deep learning approaches to predict the metric *and* the initial
momenta and show the benefits of such a DL approach (see Fig. 7). We do not use previously computed ground-truth
momenta but we backpropagate through the discretized RDMM equations and the network predicting the momenta.

**3.** *Parts of the writing that were difficult to follow.* (R2/R4). We will simplify and improve our notation, will better
explain the proposed optimal mass transport penalty, and will clarify the explanation of our learning approach.

**R1: Modeling.** *Do time-dependent velocity fields matter?* Yes, from a theoretical point of view (see 1 above).
Practically, when very large deformations (e.g., lung motions or large shape changes) are to be modeled. This might
not directly be observable in overlap measures, but will impact deformation paths. We will clarify this. *Should the
regularizer/prior not be independent from data?* Instead of putting a prior directly on a displacement, velocity, or
momentum field, *we put the prior on the metric itself.* This prior on the metric is in the reviewer's sense data independent,
but the data drives which one of the metrics is selected in the end.

**R1: Technical questions.** *How is the velocity field bounded?* Our goal is diffeomorphic registration, but there are, of
course, cases (e.g., sliding) where one would not want to use such a model. Velocity fields are bounded by predicting
the pre-weights. Weights are then computed via smoothing of these pre-weights which assures that velocity fields stay
bounded. *Symmetry loss: does invertibility imply symmetry?* RDMM aims at diffeomorphisms. Hence, we compute a
transformation (and its inverse) from source to target space. However, this does not mean that swapping source and
target images (A→B vs B→A) *also* swaps the transformations. To encourage this, we add the symmetry loss.

**R1: Experiments.** *Missing validation test/set discussion.* We use the same approach as the cross-subject experiment of
[33]: *training (2,800 pairs), validation (50 pairs) and testing (300 pairs);* all separate. We will clarify this. *Results
are similar to the baselines.* Our focus was not on outperforming current state-of-the-art registration approaches. We
agree with R1 that *our work should be considered a more theoretical contribution opening up further research* for the
estimation of spatially-temporally-varying regularizers. We will clarify this in the final manuscript.

**R1: Positioning.** *Does it fit in NeurIPS?* We think so. While we apply our model to medical registration, it is much
more general. One could easily, for example, modify our model to transform densities. This could then be used for
generative models, for example. We will make such connections more explicit.

**R2: Technical questions.** *Why are there foldings?* Our model assures diffeomorphisms in the continuum. However, as
numerical discretization will only approximate this continuum solution, foldings may occur. As the computation of the
Jacobian is also discretized, it is itself only approximately correct. We will add this to the discussion.

**R4: Technical questions.** *How are the initial momenta learned / differences to [23] / value of the DL approach?* The
equations underlying RDMM are discretized and this model is appended to the deep networks predicting the initial
momenta and pre-weights. Hence no ground-truth momenta are necessary. [23] only considers an SVF model and
uses numerical optimization to estimate SVF initial-momenta. Instead, we predict the pre-weights *and* the initial
momenta with deep networks, within this new RDMM model which allows for an estimated regularizer to move with
the deforming image. Fig. 6 compares various optimization/learning-based and spatially varying/non-varying methods.
*Fig. 7 illustrates the benefit of DL, which results in more anatomically meaningful regularizers (init_w (learn b)) than
estimation via pairwise optimization (init_w (opt b)), which is much noisier. Method will be limited by what it has seen
during training.* We agree. In many cases this is desired, as it amounts to a statistical deformation model preventing
transformations considered too unusual. We will highlight the possible limitations for capturing unusual deformations.

**R4: Experiments, references, others.** *The experiments use* $160^3$ *and* $200^3$ *images - is this a realistic?* Our approach
could scale to larger images, but we opted for slight downsampling to reduce computational requirements and GPU
memory demands. *Spatially-varying method has been studied . . . in MICCAI'13, MedIA'15.* We will include these
references. Note that in these references metric estimation is in a fixed atlas-space; whereas we address general pairwise
registration. *Is it really end-to-end training?* It is not end-to-end with respect to inclusion of the affine step. But the
non-parametric registration is (given the affine initialization). We will clarify this.

[Meta-Review · NeurIPS 2019]

All reviewers agree that this paper should be accepted. The work bridges techniques often associated with medical image analysis with those of the NeurIPS community.